# Implantation Failure: Where to Look Up?

**DOI:** 10.3390/jcm14228163

**Published:** 2025-11-18

**Authors:** Antonis Makrigiannakis, Tatjana Motrenko, Marwa Lahimer, Fanourios Ioannis Makrygiannakis, Rosalie Cabry, Jan Tesarik, Moncef Benkhalifa

**Affiliations:** 1Department of Obstetrics and Gynecology, University Hospital and School of Medicine, University of Crete, 71003 Heraklion, Greece; 2Human Reproduction Center, 85310 Budva, Montenegro; 3IVF & Reproductive Genetics, University Hospital and School of Medicine, Picardie University Jules Verne, Centre Hospital-Universitaire Amiens Sud, 80000 Amiens, France; 4PERITOX-(UMR-I 01), UPJV/INERIS, UPJV, CURS, Chemin du Thil, 80025 Amiens, France; 5MARGen Clinic, 18002 Granada, Spain; jantesarik13@gmail.com

**Keywords:** recurrent implantation failure, euploid embryo transfer, endometrial receptivity, uterine cavity, lifestyle factors

## Abstract

A competent embryo and a receptive endometrium are essential for adequate embryo–maternal cross-talk and successful implantation. However, the majority of women undergoing IVF do not achieve pregnancy after the first embryo transfer, incriminating potential implantation issues. According to statistics, recurrent implantation failure (RIF), for which different definitions have been proposed, is estimated to affect about 10–25% of women undergoing in vitro fertilization (IVF). RIF is a complex condition with overlapping causes. The primary objective of this review is to explore factors such as gamete and embryo quality, chromosomal abnormalities, uterine environment, endometrial receptivity, immune cell biomarkers, and microbiota dysregulation to better understand and overcome RIF challenges. It also highlights the significance of comprehensive evaluations of novel therapies, such as activated autologous platelet-rich plasma (PRP) or peripheral blood mononuclear cell (PBMC) insemination, on pregnancy outcomes in patients with RIF.

## 1. Introduction

Different definitions have been proposed concerning the term of recurrent implantation failure (RIF). For the ASRM (American Society of Reproductive Medicine), RIF is a condition that occurs when embryos fail to implant after multiple in vitro fertilization (IVF) attempts, while the ESHRE (European Society of Human Reproduction and Embryology) reported that RIF can be considered after failure to achieve a clinical pregnancy after the transfer of at least three good-quality embryos or two transfers of euploid embryos, when uterine and endometrial factors have been ruled out. Consequently, different IVF centers may use different definitions. However, there is no single universally accepted definition; according to Coughlan et al., 2014, RIF is defined as failure to achieve a clinical pregnancy after transfer of ≥4 good-quality embryos in a minimum of three fresh or frozen IVF cycles in women under 40 years of age [1].

For some, the term RIF stands for conditions in which clinical pregnancy is not achieved after three consecutive IVF attempts, in which one to two embryos of high-grade quality are transferred in each cycle [2]. However, no standardized criteria exist to specify the number of failed cycles or the total number of embryos transferred during these attempts [3]. Different causes of RIF, including dysregulation of molecular mechanisms governing embryo–endometrium cross-talk, the two key players in the process of implantation, were reviewed recently along with available treatment options [4].

Implantation is a process driven by a dynamic interaction between the embryo and the endometrium [5]. This “cross-talk” is a highly intricate crucial process for implantation and the success of normal placentation [6]. This interaction involves a coordinated exchange of molecular signals between a competent blastocyst and a receptive endometrium, facilitating the attachment, which is a stable adhesion of the embryo to the endometrial cells mediated by cell adhesion molecules like integrins, selectins, and cadherins [7]. The invasion, whereby trophoblast cells invade the maternal tissue to establish a connection with maternal blood vessels ensuring nutrient and oxygen supply [8], is a step forward. Dysregulation in this cross-talk can lead to implantation failure and/or complications of pregnancy.

Multifactorial RIF may result from maternal factors, including increased body mass index (BMI), anatomical disorders, smoking, advanced age that affects oocyte quality, embryo competency and endometrial receptivity [9,10], and paternal factors, including sperm maturation, competency, as well as genetic and epigenetic disorders [11].

Multiple modifiable lifestyle and environmental factors influence implantation success and ART outcomes. Strong evidence links obesity and metabolic dysfunction with impaired endometrial receptivity and reduced implantation rates [12,13]. Smoking has a well-established negative impact on uterine blood flow and gamete quality, while vaping poses emerging but likely similar risks, and its cessation should be strongly encouraged [14,15]. A dose–response meta-analysis indicated a linear association, showing that increased alcohol exposure was progressively associated with decreased fecundability; excessive intake is consistently detrimental to reproductive outcomes [16].

Moreover, evidence regarding environmental endocrine disruptors (e.g., BPA, phthalates, PFAS) is still emerging but remains biologically plausible, considering their potential effects on the female reproductive system and embryogenesis, with particular attention to associated reproductive pathologies [17].

Additionally, inadequate patient investigation, deficient management strategy, suboptimal laboratory performance, and adverse embryo transfer conditions, including embryologist and clinician skills, can all contribute to RIF [18].

In turn, the immune tolerance system can also play a critical role in implantation and pregnancy maintenance [10]. Successful implantation requires balanced uterine immunity. A one-to-one matched cohort study involving 193 patients between 2012 and 2014 revealed that uterine immune profile balance is a key factor to the success of implantation and pregnancy maintenance [19]. In addition, imbalance between the pro-inflammatory (IL-1, TNF-α) and anti-inflammatory (IL-10, TGF-β) cytokines can also be a causative factor in RIF [20]. In this context, the autoimmune factor can cause pregnancy loss, and research shows that autoantibodies targeting phospholipids can hinder implantation by causing micro thrombosis or inflammation in uterine vessels [21,22].

Abnormal uterine environment caused by different pathologies, such as anatomical abnormalities of the Mullerian tract or myomas, intrauterine adhesions, polyps, and thin endometrium, can impair embryo implantation and cause RIF [11]. These anomalies can distort the uterine cavity and increase the risk of miscarriage [18]. Among the various uterine disorders, chronic endometritis (CE), defined as a persistent inflammation of the endometrium [23], often results from intrauterine bacterial infections (such as *Mycoplasma* and *Ureaplasma*, *Chlamydia*, *Escherichia coli*, *Enterococcus faecalis*, *Streptococcus*) and can be involved in RIF [24]. On other hand, it was reported that vaginal *Lactobacillus* was significantly lower in patients with RIF compared with healthy women [25]. A systematic review and meta-analysis, involving a population of 1038 women treated between 1990 and 2024, revealed that CE seems to be linked to infertility and recurrent pregnancy loss [26].

RIF is still a complex and poorly understood area. The present review aims to find out and summarize the etiology of RIF involving gametes and embryo factors, uterine cavity environment pathophysiology, and immune tolerance balance system.

## 2. Looking at Gametes and Embryos

Oocyte quality is determined by several factors, including cytoplasmic maturation, which ensures the presence of healthy mitochondria and other resources to support early embryonic growth [27]. Nuclear maturation is equally essential for proper chromosomal alignment and segregation to minimize chromosome abnormalities and genome decay risks. Additionally, the zona pellucida must remain intact to facilitate sperm binding and protect the embryo during its initial stages. With advancing maternal age, ovaries are increasingly exposed to oxidative stress because of mitochondrial dysfunction decreasing inherent antioxidant defense (Figure 1), which significantly impacts oocyte quality and leads to an increased risk of chromosome disorders and implantation failure [28].

Similarly, sperm quality is critical to reproductive success. Sperm genome and epigenome decays are key factors in IVF outcomes, as high levels of DNA fragmentation and/or chromatin decondensation can compromise embryo quality and implantation potential (Figure 2).

Furthermore, sperm motility and morphology influence the ability of sperm to reach and fertilize the oocyte [29]. Lastly, chromatin compaction, which ensures proper DNA packaging, is essential for normal embryonic development [30]. Together, these factors determine the overall reproductive potential of the gametes and influence pregnancy outcomes. Molecular mechanisms and available treatments of both age-related and -unrelated ovarian issues were reviewed recently [31,32].

**Figure 1 jcm-14-08163-f001:**
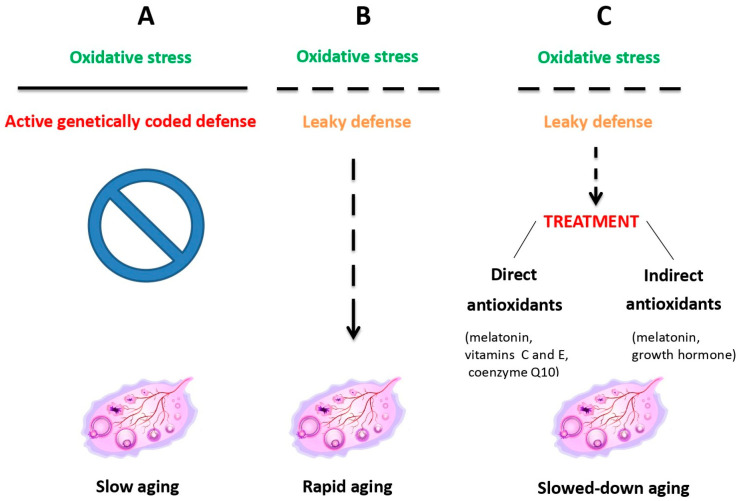
Ovarian aging due to oxidative stress resulting from mitochondrial dysfunction is counteracted by inherent antioxidant defense, which, when leaky, can be treated with external antioxidant administration. Reprinted from Ref. [31].

**Figure 2 jcm-14-08163-f002:**
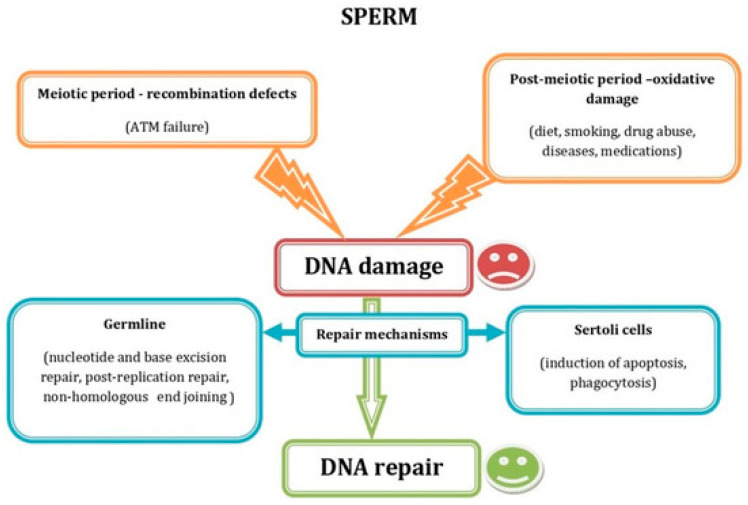
Schematic representation showing the most common factors causing sperm DNA damage during the meiotic and the postmeiotic period, as well as the DNA repair mechanisms acting in germ and Sertoli cells. Reprinted from Ref. [4].

As for the male factor, most studies to date have focused on the correlation between paternal advanced age [33,34], environmental exposure [35], and the quality of spermatozoa including WHO Criteria’s and genome integrity [36]. A review of numerous IVF/ICSI cycles, including cases involving surgically retrieved sperm, showed that severe male factors can affect fertilization rates, embryo development, and clinical outcome.

Chromosomal abnormalities in embryos play a significant role in RIF by contributing to implantation failure, miscarriage, and chromosomes and/or genes syndromes in newborns, along with specific genetic defects that can independently result in early reproductive failure [37]. Aneuploid embryos are less able to progress to the blastocyst stage or implant successfully, although euploidy can be re-established (embryo self-correction) spontaneously in some of them. In fact, aneuploidy arising at different stages of oocyte development can be repaired through inherent mechanisms acting in the oocyte, zygote, and embryo (Figure 3).

Even if implantation occurs, aneuploid embryos are more likely to result in miscarriage due to their inability to support normal fetal development [38]. Preimplantation Genetic Testing for Aneuploidy (PGT-A), known as Preimplantation Genetic Screening (PGS), is a technique used in IVF to assess the chromosomal content of embryos before transfer to the uterus. Data from patients undergoing PGT-A, often due to factors such as advanced maternal age, recurrent pregnancy loss, or multiple failed IVF cycles, highlight a high prevalence of chromosomal abnormalities in embryos and oocytes. By identifying euploid embryos, PGT-A can enhance the chances of a successful implantation, ultimately improving IVF outcomes [38] and reducing time to pregnancy. However, PGT-A also has serious limitations related to low reliability and invasiveness. To overcome this issue, replacing the conventional PGT-A with noninvasive methods for genome investigation via analysis of free DNA from spent culture media, along with the use of other noninvasive biomarkers related to chromosomal ploidy status, is currently being considered [32].

Interleukins have been found to play a crucial role in embryo development. In women undergoing IVF, higher levels of IL-6 in follicular fluid (FF) were associated with a reduction in embryo fragmentation and an increased pregnancy rate compared to those with lower IL-6 levels [39].

There is a lack of consistency in the clinical definition of RIF [40]. Most definitions currently in use are based on the number of embryos transferred with no pregnancy. However, with changing practices in embryo transfer (ET), namely from multiple to single embryos, from cleavage to blastocyst stage, and from untested to chromosomally tested embryos, the implications of a single failed ET procedure have changed [41]. A recent comprehensive survey of the definitions in use that employ this paradigm has suggested that a consensus is emerging regarding RIF as the failure to achieve a clinical pregnancy after two to three transfers with good-quality embryos and that maternal age should also be taken into account [42,43]. In a study of 105 patients who failed to achieve pregnancy after three consecutive euploid blastocyst transfers, the success rates of the fourth and fifth transfers were evaluated. Live birth rates were similar between the two (40% vs. 53.3%, *p* = 0.14). The fourth transfer also showed a comparable success rate to the first (*p* = 0.29). Overall, the cumulative live birth rate after five euploid blastocyst transfers was 98.1%, demonstrating a continued potential for pregnancy success with additional transfers [44].

A study investigated the key metabolic processes and molecular pathways throughout the preimplantation embryo development, including PI3K-Akt, mTOR, AMPK, Wnt/β-catenin, TGF-β, Notch, and Jak-Stat signaling pathways. By examining these critical molecular pathways, a review sought to highlight the differences between in vitro and in vivo embryo development and to explore the physiological and clinical implications of these differences, aiming to enhance our understanding of embryonic development and clinical practices improvement in reproductive medicine [45]. Additionally, metabolic activity can influence implantation potential, as embryos with optimal energy metabolism are better powered for sustained growth and development [46].

The ability of an embryo to implant and develop into a viable fetus is primarily determined by its intrinsic characteristics. Morphological assessment plays a crucial role, as embryos that reach the advanced blastocyst stage exhibit higher implantation rates. Cleavage stage, cell number, symmetry, and the degree of fragmentation can be assessed to help evaluate embryo quality [47].

Beyond embryo quality, the interplay between gametes also affects early embryonic events. Both oocyte and sperm quality impact cleavage rates and genome activation, while epigenetic modifications stemming from gametes can influence embryonic gene expression and implantation. However, even high-quality embryos require a receptive endometrium for implantation, highlighting the importance of embryo–endometrial dialog. Quality embryos release signaling molecules, such as cytokines, that enhance communication with the endometrium, facilitating attachment and invasion [48].

## 3. Looking at the Uterine Cavity

Implantation is a special phenomenon characterized by a dynamic cross-talk between the blastocyst and the endometrium, resulting in the formation of the placenta, which serves as the interface between the developing fetus and the maternal circulation [49]. Table 1.

Successful implantation depends on a well-coordinated interplay between the uterus, the embryo, and hormonal signals [50]. It is well known that abnormal uterus can directly or indirectly affect its ability to support embryo implantation, leading to RIF [51].

### 3.1. Uterine Anatomical Abnormalities

Congenital or acquired abnormalities of the uterus have been hypothesized to be linked with adverse fetal outcomes, pregnancy complications, and reduced fertility [51]. Various uterine abnormalities can impact implantation rates, including Müllerian anomalies, fibroids, polyps and intrauterine adhesions.

Müllerian duct anomalies (MDAs) are found in up to 7% of the general population and in nearly one-third of women with renal anomalies [52]. MDAs are hypothesized to contribute to adverse fetal outcomes, pregnancy complications, and reduced fertility [52]. Several imaging methods have been used in the assessing of MDAs, including ultrasound, hysterosalpingography, and magnetic resonance imaging (MRI) [53]. A study has demonstrated that MRI is the most effective imaging technique due to its superior capability of accurately visualizing complex uterovaginal anatomy [53].

Myomas, commonly known as uterine fibroids, are benign neoplasms of the uterine muscle that can significantly impact reproductive health [54]. In 2003, a study conducted by Baird et al. showed that the estimated prevalence of fibroids in women aged around 50 years was 70% for white women and over 80% for black women [55]. They are frequently associated with reproductive issues such as recurrent implantation failure, as they may alter uterine structure, disrupt endometrial receptivity, or impede embryo implantation [56]. Fibroids may account for 2–3% of infertility cases in women. Their impact on fertility varies based on their location in the uterus: subserosal (outside the uterus), intramural (inside the myometrium), and submucosal (inside the uterine cavity). Submucosal fibroids are associated with an increased risk of lower implantation, recurrent pregnancy loss, and less live birth rates in patients undergoing IVF [11,57]. Besides that, a septate uterus is believed to result from the incomplete resorption of the uterine septum around the 20th week of prenatal development [58]. The composition of the septum varies, ranging from a highly vascularized muscular structure to a less vascularized fibrous one, each with distinct implications for pregnancy. A more vascularized muscular septum may alter uterine motility, potentially leading to miscarriage or preterm delivery, while a less vascularized fibrous septum may impair implantation [52].

### 3.2. Intrauterine Pathology

Intrauterine adhesions, also known as Sherman’s syndrome, are formed by scar tissue within the uterus, often as a result of previous surgeries, infections, or trauma [59]. A systematic literature review, analyzing 58 articles published between 1974 and January 2022, demonstrated that this kind of adhesions can disrupt the normal uterine environment, affecting the endometrial lining and impair implantation, which may lead to infertility or recurrent pregnancy loss [60].

Beside this, endometrial polyps, considered to be benign glands, can also contribute to infertility and pregnancy complications, depending on their size and location within the uterine cavity [61]. A systematic review and meta-analysis conducted on 75 studies potentially eligible for inclusion concludes that women with endometrial polyps exhibit a higher prevalence of chronic endometritis compared to those without polyps [62]. In addition, a mini-review published in 2021 suggests that the polyps be removed in patients experiencing repeated IVF failure; however, further research is needed to confirm whether the procedure directly enhances pregnancy rates [63]. Moreover, a systematic review aims to assess the evidence on the effects of polypectomy on fecundity, implantation, and live birth rates [61]. Elias et al. (2015) [64] reported that the presence of polyps during controlled ovarian hyper-stimulation lead to an increased risk of biochemical pregnancy loss.

In the proliferative phase, the endometrium increases in thickness and becomes more vascularized, while in the secretory phase, endometrial glands grow, become tortuous, and boost their secretory activity. These changes reach their maximum about 10 to 20 days after ovulation [65]. Endocrine imbalance leads to altered oocyte maturation, uterine disorders, and endometriosis, leading to embryonic defects and decreased in vitro fertilization outcomes [17,52].

### 3.3. Intrauterine Inflammation and Chronic Endometritis

Chronic endometritis (CE) is a persistent inflammation of the endometrial mucosa, often caused by microbiological factors or mechanical and chemical irritants [66]. Currently, there is no consensus on standardized diagnostic criteria or tools for CE [67]. The prevalence of CE varies widely and is often underreported, ranging from 7.7% to 66% in cases of recurrent implantation failure and approximately 2.5% in the general infertile population. Some investigators have shown that the frequency of CE is 2.8–56.8% in infertility, 14–67.5% in recurrent implantation failure, and 9.3–67.6% in recurrent pregnancy loss [26]. From an immunological perspective, CE is characterized by the upregulation of IL-17 and the downregulation of TGFβ and IL-10 [68], and its severity is associated with a stronger Th1 and a weaker Th2 profiles [69].

CE is also more common in infertile patients with recurrent miscarriage. This condition negatively impacts the success of both spontaneous and in vitro fertilization (IVF) pregnancies and is associated with adverse perinatal outcomes, including intrauterine infections, preterm delivery, and postpartum endometritis. Diagnostic hysteroscopy should be considered for such patients to identify and manage CE effectively [70,71]. CE often presents without noticeable symptoms, making its prevalence in the general population challenging to be determined [72]. Proper investigation for CE is typically initiated only when clinical symptoms or related issues, such as infertility, arise [72].

Routinely two-dimensional ultrasonography may reveal several signs indicative of uterine abnormalities, including hematometra, hyperechogenic spots in the endometrium or along the border between the endometrium and myometrium, and varying endometrial thickness in longitudinal or transverse scans, which may suggest intracavitary synechiae [73]. Additionally, the appearance of an asynchronous endometrium with regard to the cycle phase can be observed, such as increased thickness even in the early proliferative phase, or a heteroechogenic endometrium. For patients experiencing infertility, particularly those with RIF or recurrent miscarriages, hysteroscopy should be considered as part of the diagnostic workup [74].

Histopathological evaluation involves an endometrial biopsy using a Novak curette or pipelle, with staining using hematoxylin and eosin to identify plasma cells (though less accurate), or the use of specific plasma cell stains/immunohistochemically methods such as Syndecan-1 or CD138 for a more precise diagnosis [75]. Microbiological assessment may include classical culture techniques, which are questionable and often lead to under diagnosis, or more specific molecular microbiology methods like PCR [75].

The ESHRE/ESGE CONgenital UTerine Anomalies (CONUTA) Working Group reported an initiative focused on establishing a consensus for the diagnosis of female genital anomalies. Various imaging techniques have been utilized to detect uterine malformations, each with its own potential and limitations in diagnosing the different types of malformations. These include sono-embryoscopy and magnetic resonance imaging [76].

The Cicinelli criteria, proposed in 2005, include several key features for diagnosing endometrial abnormalities: hyperemia, characterized by the expression of a vascular network at the peri-glandular level; stromal edema, resulting in a pale and thick endometrium; micro-polyps, which are small structures less than 1 mm in size; and inflammation, with the presence of inflammatory cells interspersed among normal stromal cells [77]. In 2019, Cicinelli et al. assessed the diagnostic accuracy of hysteroscopy in CE through a randomized controlled trial (RCT) observer study, aimed at evaluating the reproducibility of the proposed diagnostic criteria. The study results showed a positive impact of the criteria on physicians’ ability to identify CE [78].

Moreover, diagnosing CE is challenging for physicians. Classical histopathological staining, such as hematoxylin and eosin (HP), can identify plasma cells, but is less accurate. A more precise method involves CD138 immunohistochemistry staining or Syndecan-1 detection. Additionally, microbiological culture can be used, though it may be difficult and difficult to interpret, while PCR (Polymerase Chain Reaction) offers more specific and reliable diagnostic approach [66]. In another study, a total of 1189 hysteroscopies were performed, with biopsies and CD138 testing conducted. Among these, 735 cases (61.8%) showed no evidence of CE, while 454 cases (38.2%) were positive. CD138 testing was positive in 322 cases (27.1%). The hysteroscopic findings for CE included hyperemia in 34.7% of cases, micro-polyps in 2.1%, and interstitial edema in 3.5%. As for hysteroscopic signs in CD138-positive CE cases, the following results were observed: 17.8% had no signs, 39.5% exhibited hyperemia, 53.5% had micro-polyps, and 51.5% showed edema [79].

From the above studies, it can be concluded that infertile patients, particularly those with RIF or recurrent miscarriage (RM), exhibit a higher incidence of intrauterine pathologies, which significantly reduce the chances of achieving pregnancy, whether spontaneous or through ART. Various diagnostic modalities are available, including ultrasound, endometrial sampling (for histopathology or microbiology), and hysteroscopy. When treatment options are applicable, such as antibiotics, they should be administered based on the identified pathology. For patients who have never undergone a hysteroscopy, it is recommended as a diagnostic and therapeutic tool tailored to the specific pathology [11].

### 3.4. The Uterine Microbiota: A Key Predictor of Implantation Success

The association between uterine dysbiosis and recurrent implantation failure remains under exploration. It is a relatively recent area of investigation. Several studies have been performed to elucidate how uterine microbial imbalances might affect the ability of embryos to implant successfully during ART cycles. For many years, it was believed that a healthy fetus developed in a sterile environment [80]. However, a study published by Aagaard et al. (2014) discovered bacteria in the basal plate of the human placenta. The study outcome characterized a unique placental microbiome niche, consisting of nonpathogenic commensal microorganisms from the *Firmicutes*, *Bacteroidetes*, *Proteobacteria*, *Tenericutes*, and *Fusobacteria phyla* [81].

A study conducted in 130 women diagnosed with infertility aimed to evaluate the impact of uterine microbiota on embryo implantation success using metagenomics sequencing and microbiological examination. The population was divided into three distinct groups based on their IVF treatment history and outcome. The three groups included, respectively, women who were undergoing their first IVF attempt (*n* = 39), women who had experienced recurrent implantation failure following embryo transfer with ovarian stimulation (*n* = 27), and women who had recurrent implantation failure following frozen-thawed embryo transfer (*n* = 64). The study identified 44 species of microorganisms within the uterine cavity. Of these, 26 species were classified as opportunistic organisms, which have the potential to cause infections under certain conditions. The remaining 18 species were commensals without causing harm. These included 14 species of lactobacilli, bacteria commonly associated with maintaining a healthy microbiome, and 4 species of *Bifidobacteria*, another group of beneficial bacteria [82].

Accumulating evidence has suggested that an imbalance in the uterine microbiota could impact immune function and implantation [83], affect endometrial receptivity [84], and influence pro- and anti-inflammation balance [85]. A healthy microbiome plays a crucial role in the successful implantation of the embryo [86]. A balanced microbiome may interact directly with the embryo or with factors involved in implantation, such as the secretion of growth factors or cytokines. These interactions can help in creating a more favorable environment for the embryo’s successful attachment and development. However, when uterine dysbiosis occurs, it may lead to RIF [87,88]. According to the literature, the endometrial microbiome may act as a predictor of implantation success. The presence of specific beneficial microorganisms, especially *Lactobacilli*, is linked to a healthy endometrial environment that promotes embryo implantation. Increasing *Lactobacillus* levels to over 90% appears to enhance implantation outcomes [89,90]. *Lactobacilli* produce lactic acid, which helps maintain a low pH in the uterus, creating an environment that is inhospitable to pathogens while being conducive to implantation [91]. In contrast, pathogenic microorganisms, such as *Gardnerella vaginalis* or *Mycoplasma*, are often found in cases of dysbiosis and can contribute to a hostile uterine environment. These infections can trigger an inflammatory response that prevents proper implantation [82].

Another study, including 80 patients with RIF, compared uterine microbiome of 40 non-CE patients with that of 40 CE patients. Using Linear Discriminant Analysis (LDA), the study identified specific bacterial taxa that were characteristic of CE and non-CE patients. LDA revealed that *Proteobacteria*, *Aminicenantales*, and *Chloroflexaceae* were more commonly found in CE patients, while *Lactobacillus*, *Acinetobacter*, *Herbaspirillum*, *Ralstonia*, *Shewanella*, and *Micrococcaceae* were associated with non-CE patients. These findings highlight the distinct microbial profiles between the two groups [92]. In this context of chronic endometritis, a prospective study collected uterine endometrial specimens from 24 women with RIF, 27 RPL, and 29 fertile control women. The study found that the relative dominance rate of *Lactobacillus iners* was significantly higher in women with CE compared to women without CE. Additionally, the positive rate of *Ureaplasma* species was higher in women with CE (36.3%) than in those without CE. These results suggest that CE may play a role in the pathophysiology of both RPL and RIF, and that *Lactobacillus iners* and *Ureaplasma* species may be involved in the etiology of CE [93].

## 4. Looking at the Immune Profile

The implantation is a key event in the establishment of pregnancy. During the early stages of this process, the blastocyst embeds itself into the uterine wall [94]. This involves an interaction between the uterine epithelium and the trophoblast cells of the blastocyst, enabling attachment and invasion, both essential for the progression of pregnancy [95]. Implantation begins shortly (about 6 days) after fertilization and evolves into the formation of the placenta and the establishment of fetal–maternal interactions. This dynamic process continues to adapt and develop until approximately mid-gestation (around 22 weeks) [96]. During early pregnancy, the decidual (relating to the maternal–fetus interface) [97] immune system, such as NK cells, T cells, macrophages, and dendritic cells have specialized roles in maintaining immune tolerance to support implantation and fetal development [98]. The maternal immune system must balance tolerance and defense to allow embryo implantation [97]. The concept of immune tolerance in pregnancy refers to a healthy woman’s ability to carry a semi-allogeneic fetus without rejection by the mother’s immune system. This tolerance and the anti-inflammatory microenvironment institution are very crucial to protect the fetus and facilitate the interaction between the uterine immune cells and placental trophoblast cells [99]. The process of placental angiogenesis in pregnancy shares remarkable similarities with tumor development [100]. Like tumor cells, placental trophoblast cells are invasive, and the invasion is driven by uterine immune cells, particularly NK cells and regulatory T cells, which promote angiogenesis by secreting angiogenic factors, cytokines, and chemokines [99,100,101].

### 4.1. Immune Cells in Pregnancy

#### 4.1.1. uNK Cells

Uterine Natural Killer (uNK) cells are the most abundant immune cells in the decidua during early pregnancy. In humans, two distinct populations of circulating NK cells, the cytotoxic NK cells, characterized by CD56^dim^/CD16^+^ expression and representing approximately 90% of all circulating NK cells, and cytokine-producing NK cells, marked by CD56^bright^/CD16^−^ expression [102,103]. uNK cells regulate trophoblast invasion and blood vessel remodeling but can hinder implantation if overactive [104]. The number of uNK cells varies throughout the menstrual cycle, with their proliferation being closely linked to the influence of sex steroid hormones, particularly estrogen and progesterone [4]. These hormones play a crucial role in regulating uNK cell expansion during the late secretory phase and within the decidualized endometrium, preparing the uterine environment for potential implantation and pregnancy [105]. uNK cells becomes abundant during the mid-luteal phase of the menstrual cycle due to the influence of progesterone. This latter acts on stromal cells in the uterine lining, prompting them to produce interleukin-15 (IL-15), IL-12, and IL-18. In particular, Il-18 is a cytokine crucial for uNK cell recruitment and activity. This mechanism is important for preparing the uterine environment for potential implantation and early pregnancy [104]. Additionally, uNK cells interact with extra villous trophoblast (EVT) cells through specific interactions between killer-cell immunoglobulin-like receptors (KIRs) on the uNK cells and human leukocyte antigen (HLA) molecules expressed on EVT cells [104].

Besides that, uNK cells play a vital role in facilitating trophoblast invasion by producing key molecules like the Interferon-gamma (IFN-γ) to promote remodeling of uterine spiral arteries, creating a favorable environment for trophoblast invasion and supply to the developing fetus. Vascular endothelial growth factor (VEGF) stimulates angiogenesis to support increased blood flow to the placenta, while Granulocyte–Macrophage Colony-Stimulating Factor (GM-CSF) enhances trophoblast survival, proliferation, and invasion [106,107]. These contributions by uNK cells are critical for maintaining immune tolerance, establishing a functional placenta and ensuring successful pregnancy [108,109].

#### 4.1.2. T Cells

T cells, a type of lymphocyte, play a central role in the adaptive immune system. The T helper (Th) cell expresses CD4^+^ and displays dichotomy of Th1/Th2 immune responses. First proposed by Mosmann et al. in 1986, this concept was later adapted to explain maternal immune tolerance to fetal alloantigens [110,111]. Th1 cells drive pro-inflammatory responses by secreting cytokines like IFN-γ, TNF-α, and IL-2, which activate macrophages, enhance cytotoxic T cell activity, and promote cellular immunity. In contrast, Th2 cells are associated with anti-inflammatory and immune-tolerant responses, producing cytokines such as IL-4, IL-5, IL-10, and IL-13, which inhibit Th1-driven inflammation and support humoral immune functions [112].

The T helper Th1/Th2 theory remains a foundational concept in reproductive immunology that explains how the balance between two subsets of T helper cells, Th1, and Th2, affects immune responses [113], particularly during pregnancy. It emphasizes that a shift in the immune system’s balance is necessary to protect the semi-allogeneic fetus from maternal immune rejection while still maintaining overall immune function. This immune–microenvironment interaction also includes other T helper subsets such as Th17 cells, T cytotoxic (Tc), T regulatory (Treg), and B cells and is essential for the regulation of uterus immunity [111]. In fact, the dysregulation of Th cell immunity during pregnancy can lead to obstetrical complications, including recurrent pregnancy loss [111]. A study published by Makhseed in 2001 analyzed peripheral blood mononuclear cells from 54 women with a history of at least three normal pregnancies, 24 women with a history of recurrent spontaneous abortion (RSA), and 39 women with a history of RSA followed by normal pregnancy. The study results revealed that Th2 dominance reduces inflammation and prevents the maternal immune system from attacking the fetus. Meanwhile, excessive Th1 activity can lead to pregnancy complications such as recurrent miscarriages, preterm labor, or preeclampsia as it can trigger inflammation and rejection of the fetus [114]. Successful pregnancies are associated with a stronger Th2 bias, while spontaneous abortion and recurrent pregnancy loss are linked to a stronger Th1 bias [114].

Th17 and Treg cells play opposing but complementary roles in immune regulation, both crucial for maintaining immune homeostasis, immune tolerance and defense. This balance can be disrupted in certain situations, such as pregnancy and autoimmunity [115]. In pregnancy, the maternal immune system is challenged by fetal antigens, and Treg cells suppress Th17 cells to support fetal survival. Interestingly, autoimmune symptoms improve during pregnancy, highlighting the opposing roles of Th17 and Treg cells [116].

The study of Figueiredo and Schumacher explored the Th17/Treg ratio in both pregnancy and autoimmune conditions and suggested that understanding the balance between these cells could lead to new treatments for pregnancy loss and autoimmunity [115].

During the first trimester of pregnancy, Th17 cells, a subset of T helper cells, have been shown to play a role in the survival, proliferation, and invasion of human trophoblast cells [117]. The secretion of IL-17 by Th17 cells plays a crucial role in promoting processes essential for establishing a successful pregnancy. Th17 cells exhibit significant plasticity; while they are typically linked to inflammatory responses, they can adapt in the context of pregnancy and assume a more supportive role. Specifically, decidual Th17 cells (found in the uterine lining) appear to be associated with Th2 responses. The production of IL-4, a cytokine typically associated with Th2 cells, by decidual Th17 cells suggests that they may help create a more tolerant immune environment in the uterus, promoting fetal survival and preventing immune rejection [115,117]. Th17 and Th22 subsets, including classical and alternative forms, are distinguished from naïve CD4^+^ T cells by the cytokines that drive their differentiation. Once activated, Th17 and Th22 cells are vital for sustaining pregnancy while they also participate in the defense against extracellular pathogens at the maternal–fetal interface. Achieving the correct balance between Th1/Th2 and Treg/Th17 responses at the right time is critical for a successful pregnancy [111].

#### 4.1.3. Macrophages

Macrophages are now recognized as crucial players in embryo implantation, a key step in establishing a successful pregnancy. As part of the innate immune system, macrophages M1 and M2 [118] contribute to creating an immune-tolerant and supportive environment in the uterus, ensuring proper communication between maternal tissues and the developing embryo and thus preventing maternal rejection of the semi-allogeneic fetus. During implantation, macrophages are involved in remodeling the uterine lining (endometrium) to facilitate the invasion and anchoring of trophoblast cells. They contribute to the breakdown of extracellular matrix components, allowing the embryo to embed in the uterine wall [98]. Macrophages secrete leukemia inhibitory factors that modify the glycosylation structures of epithelial cells necessary for embryo attachment [119]. They also play a role in facilitating implantation by supporting the development of the corpus luteum and enhancing progesterone secretion [120,121,122].

A study analysis revealed that implantation failure was associated with low blood progesterone levels [123]. Additionally, the removal of M2 macrophages leads to a predominance of M1 macrophages in the uterus and elevated tumor necrosis factor-α (TNF-α) mRNA expression, triggering inflammation. Intrauterine M2 macrophages play a crucial role in preventing excessive inflammation caused by M1 macrophages, regulating the proliferation of endometrial epithelial cells, and promoting their differentiation, which collectively contribute to successful implantation.

Macrophages play a vital role in implantation and placental development through two key mechanisms, angiogenesis and inflammation modulation. They promote angiogenesis by secreting vascular endothelial growth factor (VEGF), ensuring an adequate blood supply to the developing placenta and embryo [124]. Additionally, macrophages regulate the local inflammatory response necessary for implantation, preventing it from becoming excessive or harmful [125]. By releasing cytokines and growth factors, they create a supportive microenvironment. A delicate balance between pro-inflammatory (M1) and anti-inflammatory (M2) macrophage phenotypes is essential during this process. M1 macrophages drive the initial inflammatory response, while M2 macrophages facilitate tissue repair, angiogenesis, and immune tolerance, collectively supporting successful implantation and placental development [126].

### 4.2. Evidence of Immune Disorders in RIF

Women with RIF may have alterations in the number or function of endometrial immune cells. In those with RIF, the infiltration of 14 immune cell types, including natural killer T cells, macrophages, immature and activated dendritic cells, CD56dim natural killer cells, Th1 and Th2 cells, T follicular helper cells, regulatory T cells, immature B cells, as well as various subsets of CD8 and CD4 T cells, is significantly reduced [120]. Moreover, RIF is associated with specific immunological features, including a high expression of HLA-DR on natural killer cells and HLA-F in the endometrial stroma, as well as elevated levels of both HLA-F and soluble HLA-G (sHLA-G) in the endometrial glands, suggesting a potential immunological imbalance [127]. Sfakianoudis et al., 2021 reported that elevated numbers of uNK cells, associated with their defective activity leading to cytotoxicity, are linked to RIF and recurrent miscarriages. Moreover, proposed treatments, such as glucocorticoids, intra lipids, and intravenous immunoglobulins, lack robust evidence of safety and efficacy [128].

RIF in assisted reproduction technology is a significant challenge, often linked to immune structural disorders in the endometrium. A study enrolled 42 women with RIF and compared them to fertile gestational carriers. The study results identified a unique immune profile in one-third of RIF cases, termed the “not transformed endometrial immune phenotype.” This profile includes high HLA-DR expression on NK cells, an increased fraction of CD16+ NK cells, and a reduced fraction of CD56bright NK cells. Patients with RIF also exhibited altered cytokine expression, with higher IL18/TWEAK and IL15/Fn14 ratios, reduced levels of tumor necrosis factor-like weak inducer of apoptosis, and discrepancies in IL18 mRNA expression. Immune abnormalities were present in 66.7% of cases, potentially contributing to implantation failures despite genetically tested embryo transfers [129].

Dysregulation of the immune system in RIF also involves dendritic cells (DCs) and regulatory T cells, which are integral components of this immune environment [112,130]. ILT4 (Immunoglobulin-like transcript 4) is an inhibitory receptor expressed on dendritic cells, contributing to their tolerogenic properties [127]. In fertile women, ILT4+ dendritic cells help maintain immune tolerance at the maternal–fetal interface [130]. Studies suggest that in women with RIF, ILT4+ dendritic cells are significantly downregulated. This reduction may lead to an increase in immune activation and inflammation, disrupting the immune tolerance needed for implantation [130]. Besides that, FOXP3+ cells are crucial for suppressing excessive immune responses. In fertile women, ILT4+ dendritic cells are positively associated with FOXP3+ Tregs. In RIF cases, this association is disrupted, and the downregulation of ILT4+ dendritic cells impairs the recruitment, activation, or maintenance of FOXP3+ Tregs, leading to an imbalanced immune response. This disruption may contribute to the failure of implantation or recurrent miscarriages [127,130].

In the context of RIF, immune cells, including decidual M2 macrophages at the maternal–fetal interface and the activated dendritic cells, are two key players in this immune regulation [131]. The population of decidual M2 macrophages is significantly reduced in women with RIF. This decrease leads to a diminished anti-inflammatory environment, disrupting the delicate immune balance required for embryo implantation and placental development. Similarly to M2 macrophages, activated tolerogenic dendritic cells are significantly reduced in RIF and this reduction contributes to an impaired ability to recruit and activate Tregs, leading to heightened immune activation and increased inflammation [131]. This results in increased levels of pro-inflammatory cytokines and decreased suppression of harmful immune responses, creating an unfavorable environment for embryo implantation and early pregnancy maintenance [131].

Research demonstrated the association between RIF and cytokine profile imbalance at the maternal–fetal interface. While IL-6 (pro-inflammatory cytokine) is necessary for normal physiological processes, its excessive levels can lead to chronic inflammation and immune dysregulation [132]. In addition, high levels of IL-6 may also interfere with trophoblast invasion and promote an environment unsuitable for maintaining immune tolerance [133]. Similarly, IL-10 is a potent anti-inflammatory cytokine that plays a critical role in promoting immune tolerance. The reduced IL-10 levels in RIF patients result in a diminished anti-inflammatory response. This reduction leads to an inability to counterbalance the effects of pro-inflammatory cytokines like IL-6, further disrupting the immune environment [132].

Moreover, G-CSF (Granulocyte Colony-Stimulating Factor) is a growth factor involved in modulating the immune response and enhancing trophoblast function [134]. It supports the development and function of uNK cells and promotes angiogenesis, both of which are crucial for implantation [135]. Research revealed that a deficiency in G-CSF results in impaired endometrial receptivity, reduces trophoblast invasion, and leads to an inadequate support for embryo implantation. Targeting IL-6 with specific inhibitors to reduce its pro-inflammatory effects, and administering IL-10 or G-CSF to restore the immune balance, offers a promising avenue for improving implantation outcomes in RIF patients [132].

### 4.3. Immune Checkpoints

The involvement of immune checkpoints is essential for maintaining maternal immune tolerance during pregnancy. Their main role is to prevent allogenic fetal rejection [136] by modulating the maternal immune response to recognize the fetus, which carries paternal foreign antigens [137], while simultaneously preserving immune defense against infections. These checkpoints are expressed on immune cells to regulate the strength of immune responses while preserving self-tolerance [138]. Immune checkpoints, involving cytotoxic T-lymphocyte antigen 4 (CTLA-4), programmed cell death protein 1 (PD-1), programmed cell death ligand 1 (PD-L1), T cell immunoglobulin and mucin-domain containing protein-3 (TIM-3), lymphocyte-activation gene 3 (LAG-3), and immune receptor tyrosine-based inhibitory motif (ITIM), are a group of inhibitory pathways, expressed at the maternal–fetal interface and acting in ligand–receptor fashions [138].

A review also discusses targeting of key immune cells, including T helper (TH1/TH2, TH17/Treg) and NK cells, and immune checkpoints to help maintain pregnancy and prevent complications such as miscarriage, preeclampsia, or preterm birth [139]. Preeclampsia is a severe and potentially fatal obstetric syndrome. It is strongly associated with immunological incompatibility between the mother and fetus, with genetic factors linking immune pathways to the risk of developing preeclampsia. A study by Lokki et al. investigated the immunogenic and immunomodulatory mechanisms that may contribute to the breakdown of tolerance, inflammation, and autoimmunity, all of which could lead to the development of preeclampsia [137,140].

A study by Schepanski et al., 2018 [141] discussed how prenatal challenges, integrating maternal hormones and immune markers, can influence fetal brain development and the offspring’s cognitive functions later in life. This review highlights the potential impact of maternal immune cells and cytokines on fetal brain development and the offspring’s cognitive and intellectual functioning that can persist across multiple generations reinforcing the concept of epigenetic inheritance [137,141]. Besides that, autophagy process is crucial in maintaining pregnancy by supporting fetal growth and survival. Its disruption, whether through upregulation or downregulation, can lead to pregnancy-related complications [138].

#### Limitation

To rescue RIF, it is important to improve the quality of gametes, implement adequate laboratory conditions and optimize staff skill to produce the most competent dormant blastocyst. The future challenge is how to be sure that the competent blastocyst will be able to switch via the epigenetic clock from dormant to active status and to send the appropriate signalization for the endometrium to initiate the cross-talk.

## 5. Conclusions

This review highlights modifiable factors and emerging interventions relevant to recurrent implantation failure, providing insights into mechanisms affecting implantation and endometrial receptivity. Clinically, findings emphasize the value of preconception counseling and individualized management, including weight optimization, smoking cessation, alcohol moderation, moderate physical activity, and stress reduction—interventions supported by moderate-to-strong evidence that can be readily applied in practice. While some novel therapies, such as immunomodulation or personalized embryo transfer timing, show promise, they are not yet incorporated into national or international infertility guidelines and should be considered experimental. Nevertheless, integrating evidence-based lifestyle and metabolic strategies may enhance ART outcomes and offer a practical framework for patient care.

Although significant progress has been made in understanding normal pregnancy, knowledge about RIF remains in development. Several factors have been implicated in its pathogenesis, including embryo quality, altered cytokine profiles, the involvement of checkpoint inhibitors, dysbiosis, changes in immune phenotype, and uterine abnormalities. Ongoing research continues to explore these aspects to uncover their roles in RIF and identify potential therapeutic strategies. In this context, several immune-based interventions have been proposed as supplementary treatments for RIF, based on its underlying pathophysiology. However, further research is essential to unravel the intricate interactions between endometrial immune cells. Well-designed clinical studies are necessary to validate these emerging concepts and establish their utility in clinical practice.

## Figures and Tables

**Figure 3 jcm-14-08163-f003:**
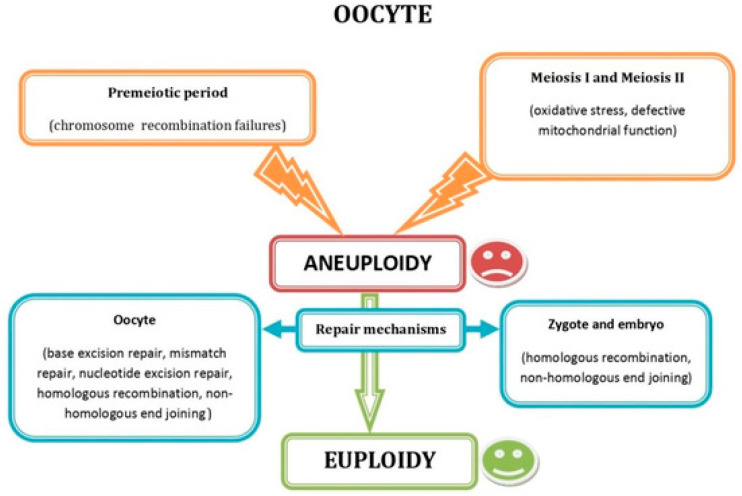
Schematic representation showing the most common factors causing oocyte aneuploidy during the premeiotic and the meiotic period, as well as the DNA repair mechanisms acting in oocytes, zygotes, and embryos. Reprinted from Ref. [4].

**Table 1 jcm-14-08163-t001:** Overview of common uterine structural abnormalities.

Category	Specific Conditions/Factors	Impact on Implantation/Fertility	Diagnostics/Assessment	Notes/Evidence
**Uterine** **anatomical** **abnormalities**	Müllerian duct anomalies (MDAs)	Reduced fertility, adverse fetal outcomes	Ultrasound, HSG, MRI (most accurate)	Found in ~7% general population; higher in women with renal anomalies
	Fibroids (myomas)—submucosal, intramural, subserosal	Submucosal: lower implantation, higher miscarriage rates, reduced live birth; intramural/subserosal: variable	Ultrasound, MRI, hysteroscopy	Fibroids account for 2–3% infertility; prevalence: 70–80% in women ~50 yrs
	Septate uterus	Impaired implantation, miscarriage, preterm delivery	Ultrasound, MRI, hysteroscopy	Vascularized muscular septum → miscarriage; fibrous septum → implantation failure
**Intrauterine pathology**	Intrauterine adhesions (Asherman’s syndrome)	Impaired endometrial environment → infertility, RIF, RPL	Hysteroscopy, ultrasound	Caused by surgery, infection, trauma; scar tissue formation disrupts implantation
	Endometrial polyps	May contribute to infertility and IVF failure	Ultrasound, hysteroscopy, biopsy	Removal suggested in repeated IVF failure; evidence on improved pregnancy rates limited
	Endocrine imbalance	Alters endometrial receptivity, oocyte maturation	Hormonal assays	Leads to embryonic defects and lower IVF outcomes
**Intrauterine inflammation/chronic endometritis (CE)**	CE	Reduces spontaneous and IVF pregnancy rates; associated with adverse perinatal outcomes	Hysteroscopy, biopsy, histology (H&E, CD138), PCR	Prevalence in RIF: 14–67.5%; immunological changes (↑ IL-17, ↓ TGFβ, IL-10, Th1/Th2 imbalance)
**Uterine microbiota**	Dysbiosis (low Lactobacillus, presence of pathogens like Gardnerella or Ureaplasma)	Negatively affects endometrial receptivity and implantation	Microbiome sequencing, culture, metagenomics	Lactobacillus >90% promotes implantation; dysbiosis linked to RIF and CE; microbial profile differs in CE vs. non-CE

## Data Availability

Not applicable.

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
