# Peer review of "Implantation Failure: Where to Look Up?"

_jcm, 2025, doi:10.3390/jcm14228163_

Round 1

Reviewer 1 Report

Comments and Suggestions for Authors

The manuscript entitled ‘implantation failure:: where to look up?’ is well written, but needs some modifications before consideration for publication.

Abstract: The abstract is not fully aligned with the manuscript. It frames both recurrent implantation failure (RIF) and recurrent pregnancy loss (RPL) as central challenges, yet the main text focuses primarily on RIF. Please either (a) restrict the abstract to RIF to match the scope, or (b) expand the manuscript to address RPL in parallel.

Keywords: The current list is too general. Consider adding more specific terms such as: recurrent implantation failure; recurrent pregnancy loss; euploid embryo transfer; endometrial receptivity; uterine cavity; lifestyle factors…

Introduction: Provide a clear, evidence-based definition of RIF (e.g., number of failed transfers and/or failed transfers of good-quality/euploid embryos) .

Extrinsic factors: Expand on modifiable lifestyle/environmental influences that may affect implantation—e.g., BMI/obesity and metabolic health, smoking/vaping, alcohol and caffeine intake, physical activity and sleep/shift work, endocrine disruptors, and severe psychosocial stress—and indicate the strength of evidence and practical counselling implications.

Uterine cavity section: This section is disproportionately long. Consider condensing the narrative and adding a concise summary table that lists each assessment.

Overall, tightening scope, aligning the abstract with the main text, expanding/clarifying the definition of RIF and extrinsic factors, and replacing lengthy prose with a synthesized table would strengthen the manuscript.

Author Response

We thank the reviewers for their positive feedback and hope that the revisions prompted by their comments have improved the manuscript.

Reviewer #1 comments on jcm-3970435:

Open Review

Quality of English Language

( ) The English could be improved to more clearly express the research.
(x) The English is fine and does not require any improvement.

Answer: We thank the reviewer for this evaluation, and we believe that it will improve our manuscript

Comments and Suggestions for Authors

The manuscript entitled ‘implantation failure:: where to look up?’ is well written, but needs some modifications before consideration for publication.

Answer: We thank the reviewer for this evaluation, and we believe that it will improve our manuscript

Abstract: The abstract is not fully aligned with the manuscript. It frames both recurrent implantation failure (RIF) and recurrent pregnancy loss (RPL) as central challenges, yet the main text focuses primarily on RIF. Please either (a) restrict the abstract to RIF to match the scope, or (b) expand the manuscript to address RPL in parallel.

Answer: We thank the reviewer for pointed out this inconsistency between the abstract and the rest of the paper. The abstract was rectified accordiang to this comment

Keywords: The current list is too general. Consider adding more specific terms such as: recurrent implantation failure; recurrent pregnancy loss; euploid embryo transfer; endometrial receptivity; uterine cavity; lifestyle factors…

 Answer: We thank the reviewer for this suggestions. We rewrote the keyword.

Introduction: Provide a clear, evidence-based definition of RIF (e.g., number of failed transfers and/or failed transfers of good-quality/euploid embryos) .

Answer: We thank the reviewer for pointed this out.

There is no single universally accepted definition, and criteria vary slightly between studies and clinical societies. Current evidence-based definitions focus on both the number of failed transfers and the quality or genetic normality of the embryos. Thats why i wanted to point this out in our manuscript.

A paragraph was added in the introduction section.

Extrinsic factors: Expand on modifiable lifestyle/environmental influences that may affect implantation—e.g., BMI/obesity and metabolic health, smoking/vaping, alcohol and caffeine intake, physical activity and sleep/shift work, endocrine disruptors, and severe psychosocial stress—and indicate the strength of evidence and practical counselling implications.

 Answer: Thank you for noticing these details. A paragraph was added : line 59_70

Uterine cavity section: This section is disproportionately long. Consider condensing the narrative and adding a concise summary table that lists each assessment.

Answer: Thank you for noticing these details, it is well appreciated. A table was added line 395

Overall, tightening scope, aligning the abstract with the main text, expanding/clarifying the definition of RIF and extrinsic factors, and replacing lengthy prose with a synthesized table would strengthen the manuscript.

Answer: We thank the reviewer for this revision, and we believe that it will improve our manuscript

Reviewer 2 Report

Comments and Suggestions for Authors

I appreciate the opportunity to review the manuscript entitled “Implantation failure: where to look up?” submitted to the journal Journal of Clinical Medicine.

Reviewer Comments:

This review addresses the challenges faced by women undergoing in vitro fertilization (IVF), with a focus on recurrent implantation failure (RIF) as well as the recurrent pregnancy loss (RPL). The authors conducted the review of a key factors linked to this issues, such as gamete and embryo quality, chromosomal abnormalities, uterine environment, endometrial receptivity, immune cell biomarkers, microbiota dysregulation etc.
However the one of the strongest aspects of this article is its exploration of novel therapeutic approaches, such as the use of activated autologous platelet-rich plasma (PRP) and peripheral blood mononuclear cell (PBMC) insemination.

The reviewer's questions:

  1. What is the scientific importance and the clinical significance of the results you showed? Are there the possibilities for introduction of the results of yours review in the clinical practice. Are these possible novel therapeutic approaches included in the guidelines of the national and/or international guidelines for the infertility treatment. Please include the appropriate paragraphs.  
  2. Please make the summaries of the all factors for the RIF/RPL analyzed in the review in the one table.

The paper is well structured and the results obtained in this research are important for the improvement of the treatment of the RIF/RPL.

Author Response

Reviewer #2 comments on jcm-3970435:

Open Review

Quality of English Language

( ) The English could be improved to more clearly express the research.
(x) The English is fine and does not require any improvement.

Answer: We appreciate the reviewer for this positive evaluation, and we believe that it will improve our manuscript

Comments and Suggestions for Authors

I appreciate the opportunity to review the manuscript entitled “Implantation failure: where to look up?” submitted to the journal Journal of Clinical Medicine.

Reviewer Comments:

This review addresses the challenges faced by women undergoing in vitro fertilization (IVF), with a focus on recurrent implantation failure (RIF) as well as the recurrent pregnancy loss (RPL). The authors conducted the review of a key factors linked to this issues, such as gamete and embryo quality, chromosomal abnormalities, uterine environment, endometrial receptivity, immune cell biomarkers, microbiota dysregulation etc.
However the one of the strongest aspects of this article is its exploration of novel therapeutic approaches, such as the use of activated autologous platelet-rich plasma (PRP) and peripheral blood mononuclear cell (PBMC) insemination.

The reviewer's questions:

  1. What is the scientific importance and the clinical significance of the results you showed? Are there the possibilities for introduction of the results of yours review in the clinical practice. Are these possible novel therapeutic approaches included in the guidelines of the national and/or international guidelines for the infertility treatment. Please include the appropriate paragraphs.  

Answer: We appreciate the reviewer for this question and suggestion, and we believe that it will enrich our manuscript. A paragraph was added in the conclusion section.

  1. Please make the summaries of the all factors for the RIF/RPL analyzed in the review in the one table.

Answer: We thank the reviewer for pointed out this information. Actually, this manuscript focuses primarily on recurrent implantation failure (RIF) rather than recurrent pregnancy loss (RPL). We are currently preparing a separate manuscript that specifically addresses RPL; therefore, we have removed sections and definitions related to RPL to maintain consistency with the scope of this paper. Similarly, the first reviewer also highlighted this point. Below are the comments from the first reviewer. « Abstract: The abstract is not fully aligned with the manuscript. It frames both recurrent implantation failure (RIF) and recurrent pregnancy loss (RPL) as central challenges, yet the main text focuses primarily on RIF. Please either (a) restrict the abstract to RIF to match the scope, or (b) expand the manuscript to address RPL in parallel. »

The paper is well structured and the results obtained in this research are important for the improvement of the treatment of the RIF/RPL.

Answer: We thank the reviewer for this positive evaluation and questions. We believe that it will enrich our manuscript.

Round 2

Reviewer 1 Report

Comments and Suggestions for Authors

The authors have responded to all requests.